# The Analysis of Selected miRNAs and Target *MDM2* Gene Expression in Oral Squamous Cell Carcinoma

**DOI:** 10.3390/biomedicines11113053

**Published:** 2023-11-14

**Authors:** Karolina Gołąbek, Dorota Hudy, Jadwiga Gaździcka, Katarzyna Miśkiewicz-Orczyk, Magdalena Nowak-Chmura, Marek Asman, Katarzyna Komosińska-Vassev, Wojciech Ścierski, Wojciech Golusiński, Maciej Misiołek, Joanna Katarzyna Strzelczyk

**Affiliations:** 1Department of Medical and Molecular Biology, Faculty of Medical Sciences in Zabrze, Medical University of Silesia in Katowice, 19 Jordana St., 41-808 Zabrze, Poland; 2Department of Otorhinolaryngology and Oncological Laryngology, Faculty of Medical Sciences in Zabrze, Medical University of Silesia in Katowice, 10 C Skłodowska St., 41-800 Zabrze, Poland; 3Department of Invertebrate Zoology and Parasitology, Institute of Biology, Pedagogical University of Cracov, Podbrzezie 3 St., 31-054 Kraków, Poland; 4Department of Clinical Chemistry and Laboratory Diagnostics, Faculty of Pharmaceutical Sciences in Sosnowiec, Medical University of Silesia in Katowice, 8 Jedności St., 41-200 Sosnowiec, Poland; 5Department of Head and Neck Surgery, Poznan University of Medical Sciences, The Greater Poland Cancer Centre, 15 Garbary St., 61-866 Poznan, Poland

**Keywords:** oral squamous cell carcinoma, OSCC, MDM2, miR-3613-3p, miR-371b-5p, miR-3658, tumour, margin

## Abstract

MiRNAs could play an important role in tumorigenesis and progression. The oncoprotein MDM2 (murine double minute 2) was identified as a negative regulator of the tumour suppressor p53. This study aims to analyse the expression of the *MDM2* target miRNA candidates (miR-3613-3p, miR-371b-5p and miR-3658) and the *MDM2* gene in oral squamous cell carcinoma tumour and margin samples and their association with the selected socio-demographic and clinicopathological characteristics. The study group consisted of 50 patients. The miRNAs and *MDM2* gene expression levels were assessed by qPCR. The expression analysis of the miRNAs showed the expression of only one of them, i.e., miR-3613-3p. We found no statistically significant differences in the miR-3613-3p expression in tumour samples compared to the margin samples. When analysing the effect of smoking on miR-3613-3p expression, we demonstrated a statistically significant difference between smokers and non-smokers. In addition, we showed an association between the miR-3613-3p expression level and some clinical parameters in tumour samples (T, N and G). Our study demonstrates that miR-3613-3p overexpression is involved in the tumour progression of OSCC. This indicates that miR-3613-3p possesses potential prognostic values.

## 1. Introduction

Oral squamous cell carcinoma (OSCC) refers to a group of malignancies affecting more than 370,000 men and women each year, accounting for an estimated 177,757 deaths in 2020 [1]. In addition, OSCC is a type of cancer with 5-year survival rates reaching only 45 to 50% [2,3]. Previous studies have shown that smoking and alcohol consumption are two well-known independent risk factors for OSCC [4,5]. In addition, genetic background, HPV (Human Papilloma Virus) infections, oral hygiene and diet are also considered risk factors [6,7,8]. Therefore, it seems important to better understand the genetic risk factors that could be crucial for developing effective diagnostic, prognostic and treatment strategies for OSCC.

MicroRNAs (miRNAs) are a class of small (about 18–22 nucleotides in length), non-coding RNAs that regulate gene expression at the post-transcriptional level by directly targeting the mRNA’s 3′ untranslated region (3′ UTR) [9,10]. These types of tissue-specific molecules could serve as oncogenes (OncomiRs) or tumour suppressors (oncosuppressor miRs) [11,12]. Accordingly, miRNAs could play an important role in tumorigenesis and progression by affecting different processes, such as cell differentiation, migration, proliferation and apoptosis [13]. miRNAs are promising key biomarkers for the diagnosis, prognosis and a possible therapeutic strategy of cancers [14].

The oncoprotein MDM2 (murine double minute 2) was identified as a negative regulator of the tumour suppressor p53. MDM2 is a protein that contains an N-terminal p53 interaction domain, central acidic domain, Zinc-finger domain and C-terminal RING domain [15]. This molecule is an E3 ubiquitin-protein ligase that binds to the tumour suppressor p53, causing its ubiquitination and subsequent proteasomal degradation [16]. The proto-oncogene *MDM2* encoding this protein is located on the human chromosomes 12q14.3 to q15 [17].

Based on several studies, it can be assumed that miRNAs directly target the *MDM2* gene to regulate tumour progression. It has also been shown that targeting the *MDM2* gene significantly reduces the viability of cancer cells and improves chemosensitivity in a p53-dependent manner [18,19,20,21,22].

The present study aims to analyse the expression of the *MDM2* target miRNA candidates (miR-3613-3p, miR-371b-5p and miR-3658) and the *MDM2* gene in oral squamous cell carcinoma tumour and margin samples and their association with the selected socio-demographic and clinicopathological characteristics.

## 2. Materials and Methods

### 2.1. Patient and Samples

The characteristics of the study group were presented in the previous study [23]. The study group comprised 50 OSCC patients recruited at the Department of Otorhinolaryngology and Oncological Laryngology in Zabrze, Medical University of Silesia, in Katowice (Poland). The tissues (paired tumour and matching margin specimens) were obtained following surgical resection. The tumour stage was assessed according to the American Joint Committee on Cancer (AJCC, version 2007) [24,25] and the WHO Classification of Head and Neck Tumours [26]. The normal tissues (margins) were checked and classified as cancer-free by pathologists. The main inclusion criteria were as follows: written informed consent to participate in the study, age over 18 years, no metabolic diseases (e.g., diabetes and hypertension), or no chronic inflammatory diseases, primary tumours and no history of preoperative radio- or chemotherapy. The data on the patients (age, sex, medical history and the use of tobacco and alcohol) were collected using an ad hoc questionnaire. The study was approved by the Bioethics Committee of the Medical University of Silesia (approval no. KNW/022/KB1/49/16 and no. KNW/002/KB1/49/II/16/17) [23]. The clinical data of the OSCC group are presented in Table 1.

### 2.2. Expression Analysis of miRNAs and the MDM2 Gene

#### 2.2.1. RNA and miRNA Extraction and Quantification

The methodology for the extraction was presented in previous studies [23,27]. All tissue samples were homogenized with the FastPrep^®^-24 homogenizer (MP Biomedicals, Solon, CA, USA) with Lysing Matrix D ceramic beads (MP Biomedicals, Solon, CA, USA). The RNA and miRNAs were extracted using the RNA isolation kit (catalogue number RIK 001, BioVendor, Brno, Czech Republic) to the standard instruction. The concentration and purity of the isolated RNA were determined using spectrophotometry in a NanoPhotometer Pearl UV/Vis Spectrophotometer (Implen, Munich, Germany) [23,27].

#### 2.2.2. Complementary DNA (cDNA) Synthesis

The methodology for the cDNA synthesis was presented in previous studies [23,27].

The obtained RNA (5 ng) was reverse-transcribed using the TaqMan^®^ Advanced miRNA cDNA Synthesis Kit (Applied Biosystems, Foster City, CA, USA) according to manufacturer’s protocol. The whole procedure consists of four reactions: poly(A) tailing reaction, the adaptor ligation reaction, the reverse transcription (RT) reaction and the miR-Amp reaction. Furthermore, the obtained RNA (5 ng) was reverse-transcribed into cDNA using the High-Capacity cDNA Reverse Transcription Kit with RNase Inhibitor (Applied Biosystems, Foster City, CA, USA), according to the manufacturer’s protocol. The reactions were prepared in Mastercycler personal (Eppendorf, Hamburg, Germany) [23,27].

#### 2.2.3. Analysis of miRNA and MDM2 Gene Expression

The methodology for the miRNA and *MDM2* gene expression analysis was presented in previous studies [23,27]. The target miRNAs of the *MDM2* gene were predicted by the miRCode (version 11) [28], miRDB (version 6.0) [29] and TargetScan (version 7.2) [30] online databases. The relative expression (RQ) of miR-3613-3p, miR-3658 and miR-371b-5p was assessed based on the guidelines using the TaqMan^®^ Advanced miRNA Assays (Assay ID: 478434_mir for miR-3613-3p; Assay ID: 478853_mir for miR-371b-5p; Assay ID: 479696_mir for miR-3658; and Assay ID: 478056_mir for miR-361-5p; Applied Biosystems, Foster City, CA, USA). The kit was supplied with primers and fluorescently marked molecular probes. All reactions were performed in the QuantStudio 5 Real-Time PCR System and Analysis Software v1.5.1 (Applied Biosystems, Foster City, CA, USA). The housekeeping miR-361-5p was used for normalizing the expression. Five surgical margin samples were used as a calibrator. RQ was calculated using 2^−∆∆Ct^ after normalization with the reference miRNA. Table 2 shows the sequences of the miRNAs.

The analysis of the relative *MDM2* gene expression (RQ) was performed by real-time PCR (qPCR) using TaqMan^®^ Gene Expression Assays (Assay ID: Hs01066930_m1 for *MDM2*; and Assay ID: Hs03929097_g1 for *GAPDH*), QuantStudio 5 Real-Time PCR System and Analysis Software v1.5.1 (Applied Biosystems, Foster City, CA, USA). The kit was supplied with primers and fluorescently marked molecular probes. The glyceraldehyde-3-phosphate dehydrogenase gene (*GAPDH*) was used as an endogenous control. Five surgical margin samples were used as a calibrator. The comparative threshold cycle (Ct) method 2^−∆∆Ct^ was used to determine the RQ [23,27].

### 2.3. HPV 16 Detection

The methodology for HPV detection was presented in a previous study [23]. DNA was extracted from tissue samples using a Gene Matrix Tissue DNA Purification Kit (EURx, Gdansk, Poland), according to the manufacturer’s instructions. The concentration and purity of the isolated DNA were prepared using spectrophotometry in a NanoPhotometer Pearl UV/Vis Spectrophotometer (Implen, Munich, Germany). HPV was detected using an AmpliSens^®^ HPV 16/18-FRT PCR kit (InterLabService, Moscow, Russia), according to the manufacturer’s protocol. All PCR reactions were performed using the QuantStudio 5 Real-Time PCR System (Applied Bio-systems, Foster City, CA, USA) [23].

### 2.4. Statistical Analyses

The Shapiro–Wilk test was used to evaluate the distribution of the variables. The median with interquartile range (25–75%) was used to describe expression of miRNAs and *MDM2* gene expression. The Mann–Whitney U test was used to compare the sociodemographic and clinical characteristics, miRNAs and the *MDM2* gene. Correlations between miRNAs and the *MDM2* gene were calculated using the Spearman’s rank correlation analysis. The level of statistical significance was set at 0.05. The statistical software STATISTICA version 13 (TIBCO Software Inc., Palo Alto, CA, USA) was used to perform all the analyses.

## 3. Results

### 3.1. miRNA Expression and Correlations between the Expression of miRNAs and Socio-Demographic and Clinicopathological Features

The expression analysis of the miRNAs showed the expression of only one of them, i.e., miR-3613-3p. We found no statistically significant differences in the miR-3613-3p expression in tumour samples compared to margin samples (*p*-value = 0.738). The median miR-3613-3p expression was 0.853 (0.515–2.424) in the tumour samples and 0.818 (0.519–1.579) in the margin samples (Figure 1).

No association was found between the miR-3613-3p expression levels, age, gender, alcohol consumption, HPV status and 3-year survival rates. When analysing the effect of smoking on the miR-3613-3p expression, we demonstrated a statistically significant difference between smokers and non-smokers (0.568 vs. 1.7845, respectively; *p*-value = 0.006) in tumour samples. In addition, we showed an association between miR-3613-3p expression level and some clinical parameters in tumour samples. Patients with T2 had a significantly lower expression level of miR-3613-3p than those with T3 (0.568 vs. 1.412, respectively; *p*-value = 0.043). Furthermore, patients with N0 had a significantly lower expression level of miR-3613-3p than those with N2 (0.591 vs. 1.257, respectively; *p*-value = 0.047). Higher miR-3613-3p expression levels were also noted in patients with G2 compared to G1 (1.581 vs. 0.455, respectively; *p*-value = 0.012). The respective results are shown in Figure 2, Figure 3, Figure 4 and Figure 5.

### 3.2. MDM2 Gene Expression and Correlations between MDM2 Gene Expression and Socio-Demographic and Clinicopathological Features

We found no statistically significant differences in *MDM2* gene expression levels in tumour samples compared to the margin samples (*p*-value = 0.075). The median *MDM2* gene expression was 0.371 (0.165–0.705) in the tumour samples and 0.555 (0.274–0.989) in the margin samples (Figure 6).

Furthermore, no association was found between the *MDM2* gene expression, age, gender, smoking, alcohol consumption, HPV status, clinical parameters (TNM and G) and 3-year survival rates in the analysed samples.

### 3.3. Correlation of the miR-3613-3p Expression and MDM2 Gene Expression

When analysing the effect of miR-3613-3p expression on *MDM2* gene expression, we showed no statistically significant differences in the tumour or margin samples (Table 3).

## 4. Discussion

MiRNAs have become the aim of many cancer studies due to their role in regulating cellular processes and facilitating the diagnosis, prognosis and treatment strategies. Using the analysis of the databases, we demonstrated that miR-3613-3p, miR-371b-5p and miR-3658 could be adequate target miRNAs of the *MDM2* gene. To the best of our knowledge, our study is the first that aimed to determine the importance of selected miRNAs and *MDM2* gene expression in OSCC tumour and margin samples and their association with the selected socio-demographic and clinicopathological characteristics. The expression analysis of the miRNAs showed the expression of only one of them, i.e., miR-3613-3p. When analysing the effect of smoking on the miR-3613-3p expression, we demonstrated a statistically significant difference between smokers and non-smokers. The effect of smoking on miRNA expression in head and neck squamous cell carcinoma has been demonstrated by other studies. The tobacco-specific nitrosamine-NNK (4-(methylnitrosamino)-1-(3-pyridyl)-1-butanone) has been identified as an important inducing factor for the upregulation of miR-21 and miR-155 and the downregulation of miR-422 [31]. Possibly, miR-3613-3p expression is also regulated by the chemical compound. In addition, we showed an association between miR-3613-3p expression level and some clinical parameters in tumour samples (T2 vs. T3, N0 vs. N2 and G1 vs. G2). Our results suggest that miRNA could be involved in the pathogenesis of OSCC. Other analyses showed the importance of miR-3613-3p in the tumorigenesis of other cancers. Chen et al. [32] reported that a frequent genomic deletion and changes in miR-3613-3p expression were found in breast cancer. The miR-3613-3p expression level was lower in the tumour tissues and serum of breast cancer patients [32]. Zhang et al. [33] demonstrated that miR-3613-3p could be an inhibitor of hepatoma cell proliferation [33]. In addition, it was discovered that miR-3613-3p was downregulated in colon cancer cell lines [34]. In their study, Pu et al. [35] found that miR-3675-3p was a promising biomarker of different stages of lung adenocarcinoma [35]. Other authors showed that miR-3613-3p regulates genes of the EGFR signalling pathway in the epithelial–mesenchymal progression of lung adenocarcinoma [36].

The roles of miR-371b-5p and miR-3658 in cancer pathogenesis have been analysed in other types of cancer (non-small cell lung cancer, hepatocellular carcinoma and bladder cancer and colorectal cancer) [37,38,39,40,41,42,43]. Luo et al. [37] found that miR-371b-5p was significantly upregulated in non-small cell lung cancer, which resulted in increased cell proliferation, migration and invasion [37]. The overexpression of this miRNA was also reported in cancers, such as hepatocellular carcinoma and bladder cancer [38,39]. On the other hand, miR-371-5p was notably downregulated in colorectal cancer and repressed the cell proliferation, invasion, metastasis and self-renewal of this cancer [40,41]. In turn, miR-3658 was upregulated in bladder cancer. Furthermore, a higher miR-3658 expression was significantly associated with lymph node invasion, distant metastasis, histological grade, TNM stage and tumour recurrence [42]. Of note, each type of cancer has a unique microRNA expression profile that could potentially distinguish it from normal tissues and other types of cancer [43].

The MDM2 protein is a key negative regulator of the tumour suppressor p53. This oncoprotein is involved in the pathogenesis of different types of cancer, such as breast cancer, cervical cancer, endometrial cancer, lung cancer, liver cancer, oesophagogastric cancer, colorectal cancer, sarcomas, osteosarcomas, gliomas, melanomas, hematopoietic malignancies, papillary thyroid cancer and ovarian cancer [16,44,45,46,47,48,49,50]. The overexpression of the MDM2 protein was also demonstrated in head and neck squamous cell carcinoma, laryngeal cancer and squamous cell carcinoma of the tonsillar region [51,52,53,54]. Friesland et al. [52] reported that changes in the level of MDM2 were also associated with a worse prognosis in the squamous cell carcinoma of the tonsil area [52]. Our analysis showed no statistically significant differences in the *MDM2* gene expression level in the tumour samples compared to the margin samples. Moreover, the change in *MDM2* gene expression was not associated with any socio-demographic or clinicopathological features.

Several miRNAs, such as miR-192, miR-194, miR-215, miR-221, miR-509-5p, miR-944, miR-1305, miR-585, miR-548b-3p, miR-219a-2-3p, miR-518, miR-143 and miR-3928, have been studied for their ability to target the *MDM2* gene. These analyses concerned various malignancies, such as multiple myeloma, hepatocellular carcinoma, pancreatic cancer, colorectal cancer, non-small cell lung cancer, glioma, breast cancer, pituitary adenoma, gastric carcinoma and bladder cancer [15,55,56,57,58,59,60,61,62,63,64]. Our study found no evidence that miR-3613-3p, miR-371b-5p and miR-3658 regulated *MDM2* gene expression in OSCC samples. Further studies on larger and more diverse groups are warranted to better understand the significance of the miRNA and its role in OSCC.

## 5. Conclusions

Our study demonstrated that miR-3613-3p overexpression is involved in the tumour progression of OSCC. This indicates that miR-3613-3p possesses potential prognostic values. Furthermore, our results reveal a differential miR-3613-3p expression in response to smoking. When analysing the effect of miR-3613-3p expression on *MDM2* gene expression, we showed no statistically significant differences in the tumour or margin samples.

## Figures and Tables

**Figure 1 biomedicines-11-03053-f001:**
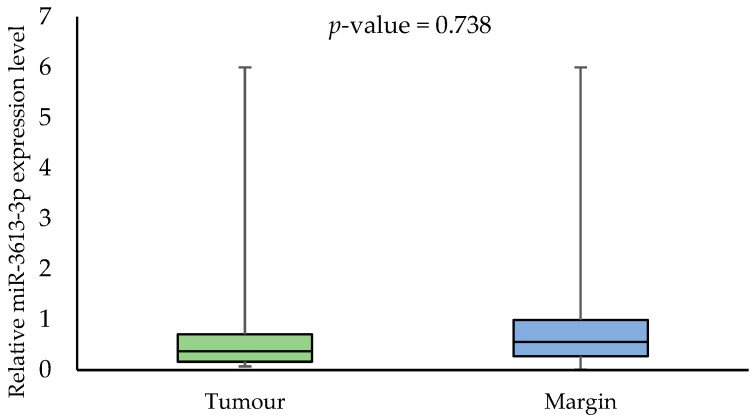
The relative miR-3613-3p expression level in tumour and margin samples.

**Figure 2 biomedicines-11-03053-f002:**
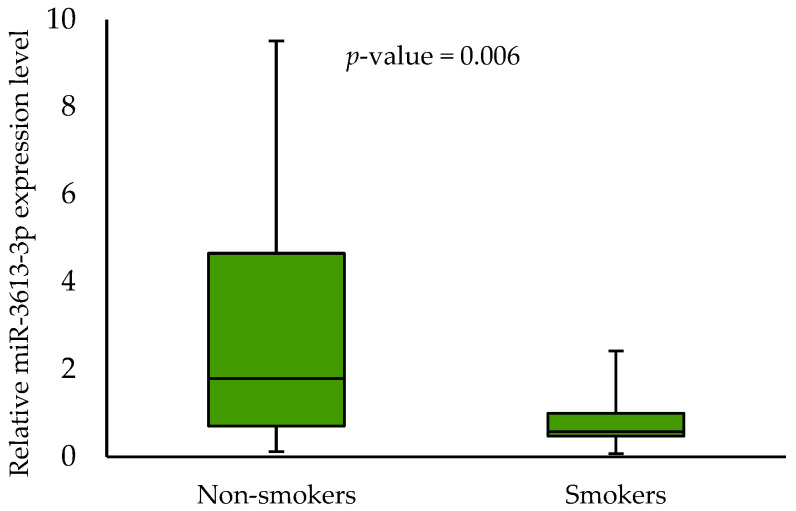
The relative miR-3613-3p expression level in non-smokers and smokers.

**Figure 3 biomedicines-11-03053-f003:**
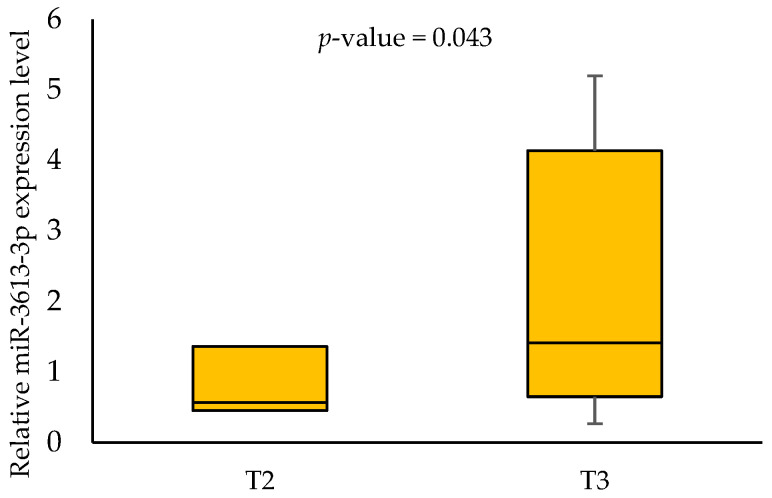
The relative miR-3613-3p expression level in patients with T2 and T3.

**Figure 4 biomedicines-11-03053-f004:**
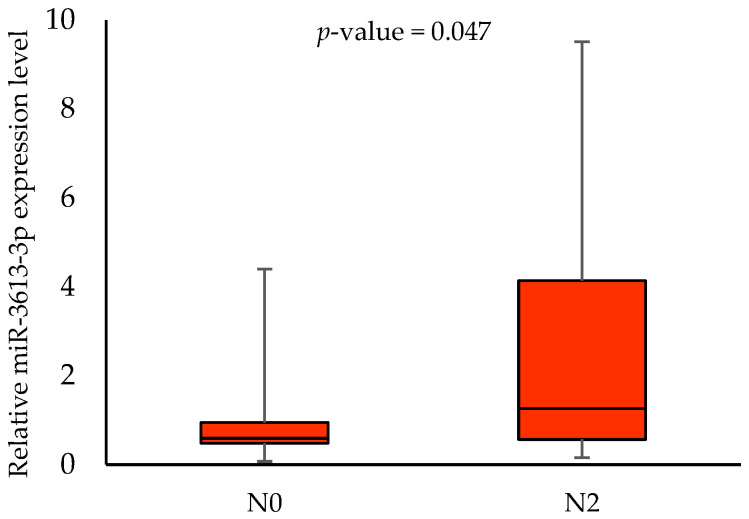
The relative miR-3613-3p expression level in patients with N0 and N2.

**Figure 5 biomedicines-11-03053-f005:**
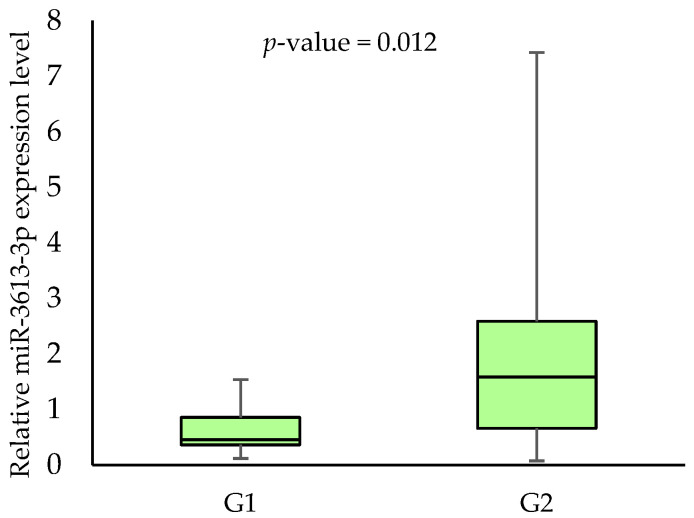
The relative miR-3613-3p expression level in patients with G1 and G2.

**Figure 6 biomedicines-11-03053-f006:**
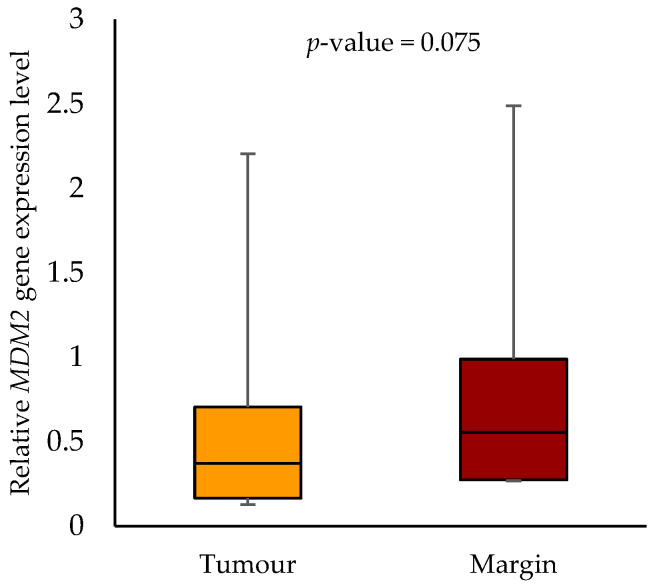
The relative *MDM2* gene expression level in the tumour and margin samples.

**Table 1 biomedicines-11-03053-t001:** Characteristics of the OSCC group.

Parameters	Patients, n (%)
Age (median): 62.5 (range: 27–87 years)	
Gender	
Men	38 (76)
Women	12 (24)
Smoking	
Smokers	28 (56)
Non-smokers	22 (44)
Alcohol consumption	
Drinker	27 (54)
Non-drinker	23 (46)
Both smokers and alcohol users	17 (34)
HPV status	
HPV-positive	13 (26)
HPV-negative	37 (74)
T classification	
T1	10 (20)
T2	23 (46)
T3	16 (32)
T4	1 (2)
Nodal status	
N0	24 (48)
N1	2 (4)
N2	20 (40)
N3	4 (8)
Histological grading	
G1	9 (18)
G2	23 (46)
G3	18 (36)
Patient status at 3 years	
Alive	12 (24)
Dead	38 (76)

**Table 2 biomedicines-11-03053-t002:** Sequences of analysed miRNAs.

miRNA	Mature miRNA Sequence
miR-3613-3p	ACAAAAAAAAAAGCCCAACCCUUC
miR-371b-5p	ACUCAAAAGAUGGCGGCACUUU
miR-3658	UUUAAGAAAACACCAUGGAGAU
miR-361-5p (Housekeeping control)	UUAUCAGAAUCUCCAGGGGUAC

**Table 3 biomedicines-11-03053-t003:** Correlation between the expressions of miR-3613-3p and *MDM2* gene in tumour and margin samples.

	Tumour	Margin
Spearman’s rank correlation coefficient		
	miR-3613-3p	
*MDM2* gene	0.68	0.49
*p*-value		
	miR-3613-3p	
*MDM2* gene	0.07	0.11

## Data Availability

The data used to support the findings of this study are available from the corresponding author upon request.

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
