# Peer review of "The Analysis of Selected miRNAs and Target MDM2 Gene Expression in Oral Squamous Cell Carcinoma"

_biomedicines, 2023, doi:10.3390/biomedicines11113053_

Round 1

Reviewer 1 Report

Comments and Suggestions for Authors

The Authors performed an analysis of the expression of the miRNAs and MDM2 gene in tumor and in margin samples and their association with the socio-demographic and clinicopathological characteristics in 50 patients affected by OSCC. The study also analysed the association of miR-3613-3p expression and MDM2 gene expression in this setting.

There are some points deserving clarification and some information that should be added.

-       - In the Abstract (page 1 line 36-37) is repetitively said that there is a correlation between the expression of miR-3613-3p and smoking.

-        -  In the Introduction (page 2 line 66) the mechanism of action of miRNA and MDM2 binding in tumorigenesis should elucidated more in depth, to make the text more understandable for readers.

-       - In Materials and Methods (page 3) the clinical characteristics (which are mentioned in the text)  should be incorporated into Table 1.

-      -  In Results (section 3.1 page 6) it might also be useful to add the graphic of the relationship between miR-3613-3p expression and smoke.

-      -  To increase the clinical utility of the manuscript, it might be useful to analyze the correlation between the expression of biological factors with outcome, in terms of tumor recurrence and treatment response.

-     -   In Discussion (page 7 lines 210-211) it was mentioned that this is the first study that analysed miRNA and MDM2 gene expression in tumor and marginal tissue in oral squamous cell carcinomas. Actually, there are other studies that analysed the expression of miRNAs in tumor and peritumoral tissue and their prognostic value. Some examples could be:

         . Ganci F et al; Altered peritumoral microRNA expression predicts head and neck cancer patients with a high risk of recurrence

         . Ganci F et al; MicroRNA expression as predictor of local recurrence risk in oral squamous cell carcinoma

         . Vahabi M et al.; MiR-96-5p targets PTEN expression affecting radio-chemosensitivity of HNSCC cells

In addition, it could also be mentioned that some studies have examined miRNA levels in OSCCs in other contexts, like the saliva, as a diagnostic and prognostic factor (Romani et al, Genome-wide study of salivary miRNAs identifies miR-423-5p as promising diagnostic and prognostic biomarker in oral squamous cell carcinoma).

-    -    In Discussion (page 8 lines 225-240) gathering the information and the citation by tumor subtype, would make the text more compact and more fluid for readers.

-    -    In the conclusion (page 8), it should be added that the study found no differences in the expression of MDM2 in the subgroups of populations analysed and in the correlations between the miR-3613-3p and MDM2 expression.

Comments on the Quality of English Language

A major English revision is needed

Reviewer 2 Report

Comments and Suggestions for Authors

The authors showed that micro RNA(miR)-3613-3p expression in oral squamous cell carcinoma tissue is associated with WHO classification. This indicates this miR holds potency as a novel predictor of oral cancer.

This manuscript has novel findings, and oral cancer clinicians will be interested in the results.

However, I have a minor point to improve this study before considering publication in our journal.

In the Methods section, the authors mentioned other miRs were assayed in this study. I don't think there was a significant difference, but it would be better to show the details of all the analysis data as a table.
